# Deconstructing Denoising Diffusion Models for Self-Supervised Learning

**Xinlei Chen[1]**   **Zhuang Liu[1,2]**   **Saining Xie[3]**   **Kaiming He[1,4]**

[1]FAIR, Meta     [2]Princeton University     [3]New York University     [4]MIT

## Abstract

In this study, we examine the representation learning abilities of Denoising Diffusion Models (DDM) that were originally purposed for image generation. Our philosophy is to *deconstruct* a DDM, gradually transforming it into a classical Denoising Autoencoder (DAE). This deconstructive process allows us to explore how various components of modern DDMs influence self-supervised representation learning. We observe that only a very few modern components are critical for learning good representations, while many others are nonessential. Our study ultimately arrives at an approach that is highly simplified and to a large extent resembles a classical DAE. We hope our study will rekindle interest in a family of classical methods within the realm of modern self-supervised learning.

## 1 Introduction

Denoising is at the core in the current trend of generative models in computer vision and other areas. Popularly known as *Denoising Diffusion Models* (DDM) today, these methods (Sohl-Dickstein et al., 2015; Song & Ermon, 2019; Song et al., 2020; Ho et al., 2020; Nichol & Dhariwal, 2021; Dhariwal & Nichol, 2021) learn a *Denoising Autoencoder* (DAE) (Vincent et al., 2008) that removes noise of multiple levels driven by a diffusion process. These methods achieve impressive image generation quality, especially for high-resolution, photo-realistic images (Rombach et al., 2022; Peebles & Xie, 2023)—in fact, these *generation* models are so good that they appear to have strong *recognition* power for understanding the visual content.

While DAE is a powerhouse of today's generative models, it was originally proposed for learning representations (Vincent et al., 2008) from data in a self-supervised manner. In today's community of representation learning, the arguably most successful variants of DAEs are based on "*masking noise*" (Vincent et al., 2008), such as predicting missing text in languages (*e.g.*, BERT (Devlin et al., 2019)) or missing patches in images (*e.g.*, Masked Autoencoder, MAE (He et al., 2022)). However, in concept, these masking-based variants remain significantly different from removing additive (*e.g.*, Gaussian) noise: while the masked tokens explicitly specify unknown *vs.* known content, no clean signal is available in the task of separating additive noise. Nevertheless, today's DDMs for generation are dominantly based on additive noise, implying that they may learn representations without explicitly marking unknown/known content.

Most recently, there has been an increasing interest (Xiang et al., 2023; Mukhopadhyay et al., 2023) in inspecting the representation learning ability of DDMs. In particular, these studies directly take *off-the-shelf* pre-trained DDMs (Ho et al., 2020; Peebles & Xie, 2023; Dhariwal & Nichol, 2021), which are originally purposed for generation, and evaluate their representation quality for recognition. They report encouraging results using these *generation-oriented* models. However, these pioneering studies obviously leave open questions: these off-the-shelf models were designed for generation, not recognition; it remains largely unclear whether the representation capability is gained by a denoising-driven process, or a diffusion-driven process.

In this work, we take a much deeper dive into the direction initialized by these recent explorations (Xiang et al., 2023; Mukhopadhyay et al., 2023). Instead of using an off-the-shelf DDM that is generation-oriented, we train models that are recognition-oriented. At the core of our philosophy is to *deconstruct* a DDM, changing it step-by-step into a classical DAE. Through this deconstructive process, we examine every single aspect (that we can think of) of a modern DDM, with the goal of

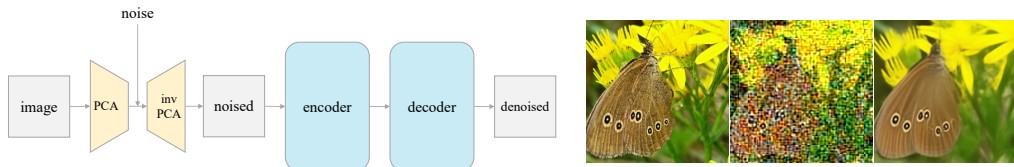

(a) latent Denoising Autoencoder (**l-DAE**)          (b) clean, noised and denoised image

Figure 1: **(a)** The **l-DAE** we have ultimately reached, after a thorough exploration of deconstructing Denoising Diffusion Models (DDM) (Ho et al., 2020), to approach classical Denoising Autoencoders (DAE) (Vincent et al., 2008). Crucial to its self-supervised representations, l-DAE adds noise in the latent space. **(b)** A clean image (left) is projected onto the latent space with PCA, in which noise is added. It is then mapped back to pixels via inverse PCA (middle). An autoencoder is learned to predict a denoised image (right).

learning representations. This research process gains us new understandings on what are the critical components for a DAE to learn good representations.

Surprisingly, we discover that the main critical component is a tokenizer (Rombach et al., 2022) that creates a *low-dimensional latent* space. Interestingly, this observation is largely *independent* of the specifics of the tokenizer—we explore a standard VAE (Kingma & Welling, 2013), a patch-wise VAE, a patch-wise AE, and a patch-wise PCA encoder. We discover that it is the *low-dimensional* latent space, rather than the tokenizer specifics, that enables a DAE to achieve good representations.

Thanks to the effectiveness of PCA, our deconstructive trajectory ultimately reaches a simple architecture that is highly similar to the classical DAE (Fig. 1). We project the image onto a latent space using patch-wise PCA, add noise, and then project it back by inverse PCA. Then we train an autoencoder to predict a denoised image. We call this "latent Denoising Autoencoder" (l-DAE).

Our deconstructive trajectory also reveals many other intriguing properties that lie between DDM and classical DAE. For one example, we discover that even using *a single noise level* (*i.e.*, not using the noise scheduling of DDM) can achieve a decent result with our l-DAE. The role of using multiple levels of noise is analogous to a form of data augmentation, which can be beneficial, but not an enabling factor. With this and other observations, we argue that the representation capability of DDM is mainly gained by the denoising-driven process, not a diffusion-driven process.

Finally, we compare with previous baselines. To make the comparisons fair and signals clean, we adopted minimal enhancements to l-DAE—only ones related to data and model size scaling. On one hand, our results are substantially better than the off-the-shelf counterparts (following the spirit of Xiang et al. (2023); Mukhopadhyay et al. (2023)): this is as expected, because these are our starting point of deconstruction. On the other hand, while l-DAE falls short in certain aspects, *e.g.* linear probing against baseline contrastive learning methods (Chen et al., 2021), or model size scaling against masking-based methods (He et al., 2022), but outshines in others (*e.g.*, fine-tuned transfer with ViT-B). These suggest opportunities and potential for further research along this direction.

## 2 RELATED WORK

In the history of machine learning and computer vision, the generation of images (or other content) has been closely intertwined with the development of unsupervised or self-supervised learning. Approaches in generation are conceptually forms of un-/self-supervised learning, where models were trained without labeled data, learning to capture the underlying distributions of the input data.

There has been a prevailing belief that the ability of a model to generate high-fidelity data is indicative of its potential for learning good representations. Generative Adversarial Networks (GAN) (Goodfellow et al., 2014), for example, have ignited broad interest in adversarial representation learning (Donahue et al., 2017; Donahue & Simonyan, 2019). Variational Autoencoders (VAE) (Kingma & Welling, 2013), originally conceptualized as generative models for approximating data distributions, have evolved to become a standard in learning localized representations ("tokens"), *e.g.*, VQVAE (Oord et al., 2017; Esser et al., 2021). Image inpainting (Bertalmio et al., 2000), essentially a form of conditional image generation, has led to a family of modern representation learning methods, including Context Encoder (Pathak et al., 2016) and MAE (He et al., 2022).

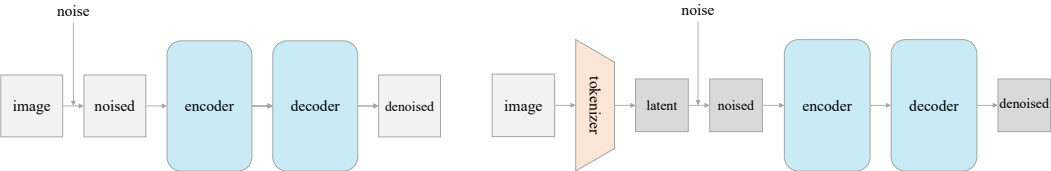

(a) a classical **Denoising Autoencoder** (DAE)  (b) a modern **Denoising Diffusion Model** (DDM)

Figure 2: **A classical DAE and a modern DDM**. **(a)** A classical DAE that adds and predicts noise on the image space. **(b)** State-of-the-art DDMs (*e.g.*, LDM (Rombach et al., 2022), DIT (Peebles & Xie, 2023)) that operate on a latent space, where the noise is added and predicted.

Analogously, the outstanding generative performance of DDMs (Sohl-Dickstein et al., 2015; Song & Ermon, 2019; Song et al., 2020; Ho et al., 2020; Dhariwal & Nichol, 2021) has drawn attention for their potential in representation learning. Pioneering studies (Xiang et al., 2023; Mukhopadhyay et al., 2023) have begun to investigate this direction by evaluating existing pre-trained DDMs. However, we note that while a model's generation capability suggests a certain level of understanding, it does not necessarily translate to representations useful for downstream tasks. Our study delves deeper into these issues, with a deconstructive trajectory different from Hudson et al. (2024).

On the other hand, although classical DAEs (Vincent et al., 2008) have laid the groundwork for autoencoding-based representation learning, their success has been mainly confined to scenarios involving masking-based corruptions (*e.g.*, He et al. (2022); Xie et al. (2022); Fang et al. (2022); Chen et al. (2023)). To the best of our knowledge, little research dives into classical DAE variants with additive Gaussian noise alone (*e.g.*, not with masking (Wei et al., 2023)). We believe the underlying reason is that a simple DAE baseline (Fig. 2(a)) performs poorly (*e.g.*, ∼20% in Fig. 4).

## 3 BACKGROUND: DENOISING DIFFUSION MODELS

Our deconstructive research starts with a Denoising Diffusion Model (DDM) (Ho et al., 2020; Dhariwal & Nichol, 2021). We briefly describe the DDM we use, following Dhariwal & Nichol (2021); Peebles & Xie (2023).

A diffusion process starts from a clean data point ($z_0$) and sequentially adds noise to it. At a specified time step $t$, the noised data $z_t$ is given by:

$$z_t = \gamma_t z_0 + \sigma_t \epsilon, \tag{1}$$

where $\epsilon \sim \mathcal{N}(0, \mathbf{I})$ is a noise map sampled from a Gaussian distribution, and $\gamma_t$ and $\sigma_t$ define the scaling factors of the signal and of the noise, respectively. By default, it is set $\gamma_t^2 + \sigma_t^2 = 1$ (Nichol & Dhariwal, 2021; Dhariwal & Nichol, 2021).

A DDM is learned to remove the noise, conditioned on the time step $t$. Unlike the original DAE (Vincent et al., 2008) that predicts a clean input, modern DDMs (Ho et al., 2020; Nichol & Dhariwal, 2021) often predict the noise $\epsilon$, minimizing:

$$\|\epsilon - \texttt{net}(z_t)\|^2, \tag{2}$$

where $\texttt{net}(z_t)$ is the network output. The model is trained for multiple noise levels given a noise *schedule* conditioned on $t$. For generation, a trained model is iteratively applied until it reaches $z_0$.

DDMs can operate on two types of input spaces. One is the image space (Dhariwal & Nichol, 2021), where the raw image $x_0$ is directly used as $z_0$. The other option is to build DDMs on a *latent* space produced by a *tokenizer*, following Rombach et al. (2022). See Fig. 2(b). In this case, a pre-trained tokenizer $f$ (which is often another autoencoder, *e.g.*, VQVAE (Oord et al., 2017)) is used to map the image $x_0$ into its latent $z_0 = f(x_0)$.

**Diffusion Transformer (DiT).** Our study begins with the Diffusion Transformer (DiT) (Peebles & Xie, 2023). We choose this for several reasons: (i) Unlike other UNet-based DDMs (Dhariwal & Nichol, 2021; Rombach et al., 2022), Transformer-based architectures can provide fairer comparisons to other self-supervised learning baselines driven by Transformers (*e.g.*, Chen et al. (2021); He et al. (2022)); (ii) DiT has a clearer distinction between the encoder and the decoder, while a UNet's

|  | acc. ($\uparrow$) | FID ($\downarrow$) |
|---|---|---|
| DiT baseline (DDAE Xiang et al. (2023)) | 57.5 | 11.6 |
| + remove class-conditioning | 62.5 | 30.9 |
| + remove VQGAN perceptual loss | 58.4 | 54.3 |
| + remove VQGAN adversarial loss | 59.0 | 75.6 |
| + replace noise schedule | 63.4 | 93.2 |

Table 1: **Reorienting DDM for self-supervised learning**. We begin with the DiT (Peebles & Xie, 2023) baseline and evaluate its linear probe accuracy (acc.) on ImageNet. This conceptually follows recent studies (Xiang et al., 2023; Mukhopadhyay et al., 2023) (more specifically, DDAE (Xiang et al., 2023)), which evaluate off-the-shelf DDMs with the same protocol. Each line is based on a modification of the immediately preceding line. The entries in gray, in which class labels are used, are not legitimate results for self-supervised learning. See Sec. 4.1 for descriptions.

encoder and decoder are connected by skip connections and may require extra efforts on network surgery when evaluating the encoder; (iii) DiT also trains much faster than other UNet-based DDMs (see Peebles & Xie (2023)) while achieving better generation quality.

Specifically, we use DiT-Large (**DiT-L**) (Peebles & Xie, 2023). In DiT-L, the encoder and decoder *together* have the size of ViT-L (Dosovitskiy et al., 2021) (24 blocks). We evaluate the representation quality of the *encoder*, which has 12 blocks, referred to as "$\frac{1}{2}$L" (half large).

**Tokenizer.** DiT instantiated in Peebles & Xie (2023) is a form of Latent Diffusion Models (LDM) (Rombach et al., 2022), which uses a VQGAN tokenizer (Esser et al., 2021). Specifically, this VQGAN tokenizer transforms the $256{\times}256{\times}3$ input image (height$\times$width$\times$channels) into a $32{\times}32{\times}4$ latent map, with a stride of 8.

**Starting baseline.** We train DiT-L for 400 epochs on ImageNet (Deng et al., 2009). The baseline results are reported in Tab. 1 (line 1). With DiT-L, we report a linear probe accuracy of **57.5**% using its $\frac{1}{2}$L encoder. The generation quality (Fréchet Inception Distance (Heusel et al., 2017), FID-50K) of this DiT-L model is **11.6**. This is the starting point of our destructive trajectory.

Despite differences in implementation details (see Appendix A), our starting point largely follows recent studies (Xiang et al., 2023; Mukhopadhyay et al., 2023), which evaluate DDMs off-the-shelf with linear probing.

## 4 Deconstructing Denoising Diffusion Models

Our deconstruction trajectory is divided into *three* stages. We first adapt the generation-focused settings in DiT to be more oriented toward self-supervised learning (Sec. 4.1). Next, we deconstruct and simplify the tokenizer step by step (Sec. 4.2). Finally, we attempt to reverse as many DDM-motivated designs as possible, pushing the models towards a classical DAE (Vincent et al., 2008) (Sec. 4.3).

### 4.1 Reorienting DDM for Self-supervised Learning

While a DDM is conceptually a form of DAE, it was originally developed for image generation. Many designs in a DDM are oriented toward the generation task. Some designs are *not legitimate* for self-supervised learning (*e.g.*, involving class labels); some others are not necessary if visual quality is not concerned. We thus first reorient our baseline for self-supervised learning (Tab. 1).

**Remove class-conditioning.** A high-quality DDM is often trained with conditioning on *class labels*, which can largely improve the generation quality. But the usage of class labels is simply not legitimate in self-supervised learning. As the first step, we remove class-conditioning in our baseline.

Surprisingly, this change substantially improves the linear probe accuracy from 57.5% to 62.1% (Tab. 1), even though generation is greatly hurt as expected (FID from 11.6 to 34.2). We hypothesize that directly conditioning the model on class labels may reduce the model's demands on encoding the label-related information. Removing this conditioning can force the model to learn more semantics.

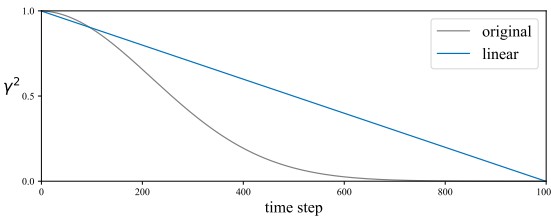

Figure 3: **Noise schedule comparisons**. The original schedule (Ho et al., 2020; Peebles & Xie, 2023), which sets $\gamma_t^2 = \Pi_{s=1}^t (1 - \beta_s)$ with a linear schedule of $\beta$, spends many time steps on very noisy images (small $\gamma$). Instead, we use a simple schedule that is linear on $\gamma^2$, which provides less noisy images.

**Deconstruct VQGAN.** The VQGAN tokenizer inherited from LDM (Rombach et al., 2022) is trained with multiple loss terms: (i) AE reconstruction loss; (ii) KL-divergence regularization loss (Rombach et al., 2022);[1] (iii) perceptual loss (Zhang et al., 2018b); and (iv) adversarial loss (Goodfellow et al., 2014; Esser et al., 2021) with a discriminator. We ablate the latter two terms next.

The **perceptual loss** (Zhang et al., 2018b) is based on a *supervised* VGG net (Simonyan & Zisserman, 2015) trained for ImageNet classification. Therefore, using the VQGAN trained with this loss is again not legitimate. Instead, we train another VQGAN (Rombach et al., 2022) where this loss is removed. Using this tokenizer *reduces* the linear probe accuracy significantly from 62.5% to 58.4% (Tab. 1), which, however, provides the first legitimate entry thus far. This comparison reveals that *a tokenizer trained with the perceptual loss (with class labels) in itself provides semantic representations*. From now on, we will no longer use perceptual loss.

We train the next VQGAN tokenizer that further removes the **adversarial loss**. It slightly increases the accuracy from 58.4% to 59.0% (Tab. 1). At this point, our tokenizer is essentially a VAE, which we move on to deconstruct next. We also note that removing either loss harms generation quality.

**Replace noise schedule.** In the task of generation, the goal is to progressively turn a noise map into an image. As a result, the original noise schedule allocates many time steps on very noisy images (Fig. 3). This is not necessary if our model is not generation-oriented.

We study a simpler noise schedule for the purpose of self-supervised learning. Specifically, we let $\gamma_t^2$ linearly decay in the range of $1 > \gamma_t^2 \geq 0$ (Fig. 3). This allows the model to spend more capacity on cleaner images. This change greatly improves the linear probe accuracy from 59.0% to 63.4% (Tab. 1), suggesting that the original schedule is too focused on noisier regimes. On the other hand, as expected, it further hurts generation, leading to a FID of 93.2.

**Summary.** Overall, the results in Tab. 1 reveal that *self-supervised learning performance is not correlated to generation quality*. The representation capability of a DDM is not necessarily the outcome of its generation capability.

## 4.2 DECONSTRUCTING THE TOKENIZER

Next, we further deconstruct the VAE tokenizer by making substantial simplifications. We compare the following four variants as tokenizers, each of which is a simplified version of the preceding one:

- **Convolutional VAE.** Our deconstruction thus far leads us to a VAE tokenizer. As common practice (Kingma & Welling, 2013; Rombach et al., 2022), the encoder $f(\cdot)$ and decoder $g(\cdot)$ of this VAE are deep convolutional (conv) neural networks (LeCun et al., 1989). This convolutional VAE minimizes the following loss function:

$$\|x - g(f(x))\|^2 + \mathbb{KL}\left[f(x)|\mathcal{N}\right].$$

  Here, $x$ is the full input image. The first term is the reconstruction loss, and the second term is the K-L divergence (Bishop & Nasrabadi, 2006; Esser et al., 2021) between the latent distribution of $f(x)$ and a unit Gaussian distribution.

- **Patch-wise VAE.** Next we simplify so that the VAE encoder and decoder are both *linear* projections, and the VAE input $x$ is a *patch*. The training process of this *patch-wise VAE* minimizes this loss:

$$\|x - U^T V x\|^2 + \mathbb{KL}\left[V x|\mathcal{N}\right].$$

---

[1]The KL form in Rombach et al. (2022) does not perform explicit vector quantization (VQ).

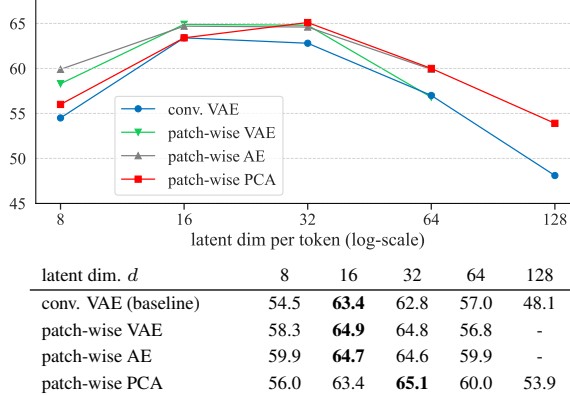

Table 2: **Linear probe accuracy vs. latent dimension**. With a DiT model, we study four variants of tokenizers for computing the latent space. We vary the dimensionality $d$ (*per token*) of the latent space. The table is visualized by the plot above. **All four variants of the tokenizers exhibit similar trends**, despite their differences in architectures and loss functions. The 63.4% entry of "conv. VAE" is the same entry as the last line in Tab. 1.

| latent dim. $d$ | 8 | 16 | 32 | 64 | 128 |
|---|---|---|---|---|---|
| conv. VAE (baseline) | 54.5 | **63.4** | 62.8 | 57.0 | 48.1 |
| patch-wise VAE | 58.3 | **64.9** | 64.8 | 56.8 | - |
| patch-wise AE | 59.9 | **64.7** | 64.6 | 59.9 | - |
| patch-wise PCA | 56.0 | 63.4 | **65.1** | 60.0 | 53.9 |

Here $x$ denotes a patch flattened into a $D$-dimensional vector. Both $U$ and $V$ are $d{\times}D$ matrices, where $d$ is the latent space dimension. Patch size is $16{\times}16$, following Dosovitskiy et al. (2021).

- **Patch-wise AE.** We further simplify VAE by dropping regularization:

$$\|x - U^T V x\|^2.$$

As such, this tokenizer is essentially an AE on patches, with a linear encoder and decoder.

- **Patch-wise PCA.** Finally, we consider a simpler variant which performs PCA on the patch space. It is easy to show that PCA is equivalent to a special case of AE:

$$\|x - V^T V x\|^2.$$

Here $V$ satisfies $VV^T{=}I$ ($d{\times}d$ identity matrix). The PCA bases can be simply computed by eigen-decomposition on a large set of randomly sampled patches without gradient-based training.

Tab. 2 summarizes the linear probe accuracy of DiT using these four variants of tokenizers. We show the results w.r.t. the *latent dimension "per token"*.[2] We draw the following observations.

*Latent dimension of the tokenizer is crucial for DDM to work well in self-supervised learning.*

As shown in Tab. 2, all four variants of tokenizers exhibit similar trends, despite their differences in architectures and loss functions.[3] Interestingly, the optimal dimension is relatively low ($d$ is 16 or 32), even though the full dimension per patch is much higher ($16{\times}16{\times}3{=}768$).

Surprisingly, the convolutional VAE tokenizer is neither necessary nor favorable; instead, all patch-based tokenizers, in which each patch is encoded *independently*, perform similarly with each other and consistently outperform conv. VAE. In addition, the KL regularization term is *unnecessary*, as both the AE and PCA variants work well.

To our further surprise, *even the PCA tokenizer works well*. Unlike the VAE or AE counterparts, PCA does *not* require gradient-based training. With pre-computed PCA bases, the application of the PCA tokenizer is close to a "image pre-processing" step, rather than a "network architecture". Its effectiveness largely helps us push the modern DDM towards a classical DAE, as we will show next.

*High-resolution, pixel-based DDMs are inferior for self-supervised learning.*

Before we move on, we report an extra ablation that is consistent with the aforementioned observation. Specifically, we consider a "naïve tokenizer" that performs *identity mapping* on patches extracted from resized images. In this case, a "token" is the flatten vector consisting all *pixels* of a patch.

In Fig. 4, we show the results of this "pixel-based" tokenizer, operated on an image size of 256, 128, 64, and 32, respectively with a patch size of 16, 8, 4, 2. The "latent" dimensions of these tokenized spaces are 768, 192, 48, and 12 per token. In all case, the sequence length of the Transformer is kept unchanged (256).

---

[2]For patch-wise VAE/AE/PCA (patch stride is 16), we treat each patch as a token, so the latent dimension is simply $d$ for each patch. For the convolutional VAE that has a stride of 8, DiT (Peebles & Xie, 2023) treats each $2{\times}2$ patch on the latent space as a "token"; as a result, its latent dimension "per token" should be multiplied by 4 for calibration.

[3]For the patch-wise tokenizers, we find they also learn different filters. See Appendix B for visualizations.

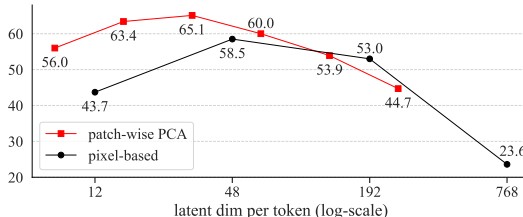

Figure 4: Linear probe accuracies on **pixel-based tokenizer**, operated on an image size of 256, 128, 64, and 32, respectively with a patch size of 16, 8, 4, 2. The "latent" dimensions of these tokenized spaces are 768, 192, 48, and 12 per token. Similar to others, this pixel-based tokenizer exhibits a similar trend: a relatively small dimension of the latent space is optimal.

Interestingly, this *pixel-based* tokenizer exhibits a similar trend with previous tokenizers, although the optimal dimension is shifted. In particular, the best dimension is $d$=48, which corresponds to an image size of 64 with a patch size of 4. With image size 256 and patch size 16 ($d$=768), the accuracy drops dramatically to 23.6%.

These comparisons show that the tokenizer and the resulting latent space are crucial for DDM/DAE to work competitively in the self-supervised learning scenario. In particular, applying a classical DAE with additive Gaussian noise on the image space leads to poor results.

### 4.3 TOWARD CLASSICAL DENOISING AUTOENCODERS

Next, we go on with our deconstruction trajectory and aim to get as close as possible to the classical DAE (Vincent et al., 2008). We attempt to remove every single aspect that still remains different between our current PCA-based DDM and the classical DAE practice. Via this deconstructive process, we gain better understandings on how every modern design may influence the classical DAE. Tab. 3 lists the results.

**Predict clean data (rather than noise).** While modern DDMs commonly predict the noise $\epsilon$ (see Eq. (2)), the classical DAE predicts the clean data instead. We examine this difference by minimizing:

$$\lambda_t \|z_0 - \texttt{net}(z_t)\|^2. \tag{3}$$

Here $z_0$ is the clean data (in the latent space), and $\texttt{net}(z_t)$ is the network prediction. $\lambda_t$ is a $t$-dependent loss weight, introduced to balance the contribution of different noise levels (Salimans & Ho, 2022). It is suggested to set $\lambda_t = \gamma_t^2/\sigma_t^2$ as per Salimans & Ho (2022). We find that setting $\lambda_t = \gamma_t^2$ works better in our scenario. Intuitively, it simply puts more weight to the loss terms of the *cleaner* data (larger $\gamma_t$).

With the change of predicting clean data, the linear probe accuracy *degrades* from 65.1% to 62.4% (Tab. 3). This suggests that the choice of the prediction target influences the representation quality.

Even though we suffer from a degradation in this step, we will stick to this modification from now on, as our goal is to get as close as possible to a classical DAE.[4]

**Remove input scaling.** In modern DDMs, the input is scaled by a factor of $\gamma_t$ (Eq. (1)). This is not common in a classical DAE and next we remove it, *i.e.*, set $\gamma_t \equiv 1$. As $\gamma_t$ is fixed, we need to define a noise schedule directly on $\sigma_t$. We simply set as a linear schedule on $\sigma_t$ from 0 to $\sqrt{2}$, and empirically set the weight in Eq. (3) as $\lambda_t = 1/(1 + \sigma_t^2)$. This again puts more emphasis on cleaner data.

After fixing $\gamma_t \equiv 1$, we achieve a decent accuracy of 63.6% (Tab. 3), which compares favorably with the varying $\gamma_t$ counterpart's 62.4%.

**Operate on the *image* space with inverse PCA.** So far, for all entries we have explored (except Fig. 4), the model operates on the latent space produced by a tokenizer (Fig. 2 (b)). Ideally, we hope our DAE can work directly on the *image* space while still having good accuracy. With the usage of PCA, we can achieve this goal by inverse PCA.

The idea is illustrated in Fig. 1. Specially, we project the input image into the latent space by the PCA bases (*i.e.*, $V$), add noise in the latent space, and project the noisy latent back to the image space by the *inverse* PCA bases ($V^T$). Fig. 1b (middle) shows an example image with noise added in the latent space. With this noisy image as the input to the network, we can apply a standard ViT network (Dosovitskiy et al., 2021) that directly operate on images, as if there is no tokenizer.

---

[4]We have revisited undoing this change in our final entry, in which we have not observed this degradation.

|                                                | acc. |
| :--------------------------------------------- | ---: |
| patch-wise PCA baseline                        | 65.1 |
| + predict clean data (rather than noise)       | 62.4 |
| + remove input scaling (fix $\gamma_t \equiv 1$) | 63.6 |
| + operate on image input with inv. PCA         | 63.6 |
| + operate on image output with inv. PCA        | 63.9 |
| + predict original image                       | 64.5 |

Table 3: **Moving toward a classical DAE** from our patch-wise PCA tokenizer. Each line adds a modification to the immediately preceding line. See Sec. 4.3 for details.

Applying this modification on the input side (while still predicting the output on the latent space) has 63.6% accuracy (Tab. 3). Further applying it to the output side (*i.e.*, predicting the output on the image space with inverse PCA) has 63.9% accuracy. Both results show that operating on the image space with inverse PCA can achieve similar results as operating on the latent space.

**Predict original image.** While inverse PCA can produce a prediction target in the image space, this target is *not* the original image. This is because PCA is a *lossy* encoder for any reduced dimension $d$. In contrast, it is a more natural solution to predict the original image directly.

When we let the network predict the original image, the "noise" includes two parts: (i) the additive Gaussian noise, whose intrinsic dimension is $d$, and (ii) the PCA reconstruction error, whose intrinsic dimension is $D - d$ ($D$ is 768). We weight the loss of both parts differently.

Formally, with the clean original image $x_0$ and network prediction $\texttt{net}(x_t)$, we can compute the residue $r$ projected onto the full PCA space: $r \triangleq V(x_0 - \texttt{net}(x_t))$. Here $V$ is the $D$-by-$D$ matrix representing the full PCA bases. Then we minimize $\lambda_t \sum_{i=1}^{D} w_i r_i^2$, where $i$ denotes the $i$-th dimension of the vector $r$. The per-dimension weight $w_i$ is 1 for $i \leq d$, and 0.1 for $d < i \leq D$. Intuitively, $w_i$ down-weights the loss of the PCA reconstruction error. With this formulation, predicting the original image achieves 64.5% linear probe accuracy (Tab. 3).

Conceptually, this variant is very simple: its input is a noisy image whose noise is added in the PCA space, its prediction is the original clean image (Fig. 1).

**Single noise level.** Lastly, out of curiosity, we further study a variant with *single-level* noise. We note that multi-level noise, given by noise scheduling, is a property motived by the diffusion process in DDMs; it is conceptually unnecessary in a classical DAE.

We fix the noise level $\sigma$ as a constant ($\sqrt{1/3}$). Using this single-level noise achieves decent accuracy of 61.5%, a 3% degradation *vs.* the multi-level noise counterpart (64.5%). This mild degeneration suggests multiple levels of noise is analogous to a form of data augmentation in DAE: it is beneficial, but not an enabling factor. This also implies that the representation capability of DDM is mainly gained by the denoising-driven process, not a diffusion-driven process.

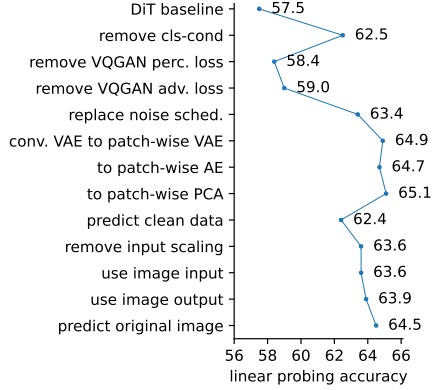

Figure 5: **The overall deconstructive trajectory** from a modern DDM to *l*-DAE, summarizing Tab. 1, Tab. 2, and Tab. 3. Each line is based on a modification of the immediately preceding line.

As multi-level noise is useful and conceptually simple, we keep it for our final comparisons.

## 4.4 SUMMARY

In sum, we have deconstructed a modern DDM and pushed it towards a classical DAE (Fig. 5). We *undo* many of the modern designs and retain only two designs inherited from modern DDMs: (i) a low-dimensional latent space in which noise is added; and (ii) multi-level noise.

We use the entry at the end of Tab. 3 as our final DAE instantiation (illustrated in Fig. 1). We refer to this method as "*latent* Denoising Autoencoder", or ***l*-DAE**.

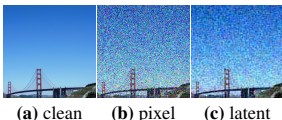

**(a)** clean    **(b)** pixel    **(c)** latent

Figure 6: **Visualizing pixel noise *vs*. latent noise**. **(a)** clean image, 256×256. **(b)** noise added to the image space. **(c)** noise added to the PCA latent space, visualized by back projection to the image space using inverse PCA. $\sigma = \sqrt{1/3}$ in both cases.

## 5 ANALYSES AND COMPARISONS

**Visualizing latent noise.** Conceptually, *l*-DAE is a form of DAE that learns to remove noise added to the latent space. Thanks to the simplicity of PCA, we can easily visualize the latent noise by inverse PCA. Fig. 6 compares the noise added to pixels *vs*. to the latent. Unlike the pixel noise, the latent noise is largely independent of the *resolution* of the image. With patch-wise PCA as the tokenizer, the pattern of the latent noise is mainly determined by the patch size. Intuitively, we may think of it as using patches, rather than pixels, to resolve the image. This behavior resembles MAE (He et al., 2022), which masks out patches instead of individual pixels.

**Denoising results.** Fig. 7 shows examples of denoising results based on *l*-DAE. Our method produces reasonable predictions despite the heavy noise. We note that this is less of a surprise, because neural network-based image restoration (Burger et al., 2012; Dong et al., 2014) has been intensively studied.

Nevertheless, the visualization may help us better understand how *l*-DAE can learn good representations. The heavy noise added to the latent space creates a challenging *pretext* task for the model to solve. It is nontrivial (even for human beings) to predict the content based on one or a few noisy patches locally; the model is forced to learn higher-level, more holistic semantics to make sense of the underlying objects and scenes.

Beyond qualitative visualizations, we perform three more quantitative analyses on *l*-DAE:

**Data augmentation.** All models we present thus far have *no data augmentation*: only the center crops of images are used, with no random resizing or color jittering, following Dhariwal & Nichol (2021); Peebles & Xie (2023). We further explore a mild data augmentation (random resized crop) for our final *l*-DAE, which has slight improvement (Tab. 4a). This suggests that *the representation learning ability of l-DAE is largely independent of its reliance on data augmentation.* A similar behavior was observed in MAE (He et al., 2022), which sharply differs from the behavior of contrastive learning methods (*e.g*., Chen et al. (2020)).

**Training epochs.** All our experiments thus far are based on 400-epoch training. Following MAE (He et al., 2022), we also study training for 800 and 1600 epochs (Tab. 4b). As a reference, MAE has a

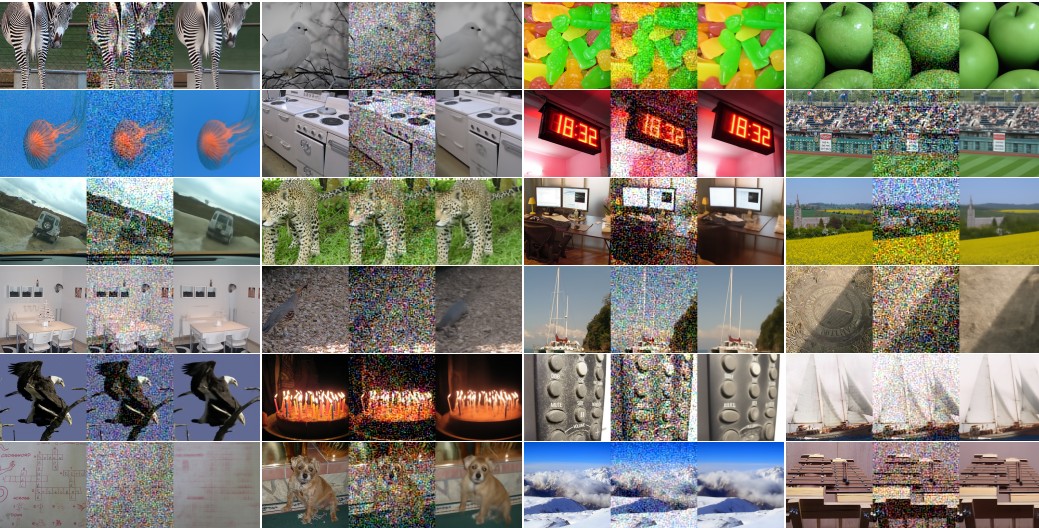

Figure 7: **Denoising results** of *l*-DAE, evaluated on ImageNet validation images. *This denoising problem, serving as a pretext task, encourages the network to learn meaningful representations in a self-supervised manner.* For each case, we show: (**left**) clean image; (**middle**) noisy image as the input to the network, where the noise is added to the latent space; (**right**) denoised output.

| aug. | center crop | random crop |
|------|-------------|-------------|
| acc. | 64.5 | 65.0 |

(a) augmentation

| epochs | 400 | 800 | 1600 |
|--------|-----|-----|------|
| acc. | 65.0 | 67.5 | 69.6 |

(b) training epochs

| encoder | ViT-B | ViT-$\frac{1}{2}$L | ViT-L |
|---------|-------|--------|-------|
| acc. | 60.3 | 65.0 | 70.9 |

(c) model size

Table 4: **Analysis** on *l*-DAE. Specifically, we show: *l*-DAE requires minimal data augmentation in **(a)** and scales reasonably well with training epochs in **(b)** and model size in **(c)**.

| acc. | ViT-B | ViT-L |
|------|-------|-------|
| MoCo v3 | **76.7** | **77.6** |
| MAE | 68.0 | 75.8 |
| *l*-DAE | 66.6 | 75.0 |

(a) linear probing

| acc. | ViT-B | ViT-L |
|------|-------|-------|
| MoCo v3 | 83.2 | 84.1 |
| MAE | 83.6 | **85.9** |
| *l*-DAE | **83.7** | 84.7 |

(b) end-to-end fine-tuning

| $AP^{box}$ | ViT-B | ViT-L |
|------------|-------|-------|
| Supervised | 47.6 | 49.6 |
| MAE | 51.2 | **54.6** |
| *l*-DAE | **51.6** | 54.4 |

(c) object detection transfer

Table 5: **System-level comparisons with previous baselines** on standard Transformer encoders. For ImageNet classification, we compare MoCo v3 (Chen et al., 2021) and MAE (He et al., 2022) in two protocols: **(a)** linear probing and **(b)** End-to-end fine-tuning. For object detection transfer **(c)**, we use ViTDet (Li et al., 2022) and report AP on COCO (Lin et al., 2014). Different methods show different strengths: *l*-DAE shines in fine-tuned transfer with ViT-B; and even for linear probing, the accuracy gap is *drastically* closed (from ~20% in Fig. 4 to 70+% in Tab. 5a).

significant gain (4%) from 400 to 800 epochs, and MoCo v3 (Chen et al., 2021) has nearly no gain (0.2%) from 300 to 600 epochs. Our *l*-DAE falls in-between (+4.6% from 400 to 1600 epochs).

**Model size.** So far, our all models are based on the DiT-L variant (Peebles & Xie, 2023), whose encoder and decoder are both "ViT-$\frac{1}{2}$L" (half depth of ViT-L). We further train models of different sizes, whose encoder is ViT-B or ViT-L (decoder always has the same size). We observe a good *scaling* behavior w.r.t. model size (Tab. 4c): from ViT-B to ViT-L it has a large, 10.6% gain. A similar scaling behavior is also observed in MAE (He et al., 2022), which gains 7.8% from ViT-B to ViT-L.

**Comparison with previous baselines on ImageNet.** We compare *l*-DAE to previous baselines in Tab. 5a and Tab. 5b. We consider MoCo v3 (Chen et al., 2021), which belongs to contrastive learning methods, and MAE (He et al., 2022), which belongs to masking-based methods. Besides linear probing, we also report end-to-end fine-tuning results, with details listed in Appendix A.

While *l*-DAE falls short in linear probe accuracy especially against MoCo v3, it behaves similarly to MAE, and even achieves the *best* overall fine-tuning results with ViT-B. We note that here the training settings are made *as fair as possible* between MAE and *l*-DAE: both are trained for 1600 epochs and with random crop augmentation. With our deconstructive efforts, *the accuracy gap between MAE and a DAE-driven method is drastically closed (from ~20% in Fig. 4 to 70+% in Tab. 5a)*.

**Comparison on object detection transfer.** Finally, we transfer the representations learned with *l*-DAE for object detection and segmentation. We use the ViTDet framework (Li et al., 2022) and evaluate on the COCO benchmark (Lin et al., 2014) against MAE and supervised pre-training. Again, we find *l*-DAE behaves similarly to MAE, and significantly outperforms supervised pre-training. With MAE, *l*-DAE performs well with ViT-B, while slightly falls short when the model size gets larger. More details and segmentation results are found in Appendix A and Appendix C.

## 6  CONCLUSION

We have reported that *l*-DAE, which largely resembles the classical DAE, can perform competitively in self-supervised learning. The critical component is a low-dimensional latent space on which noise is added. We hope our discovery will reignite interest in denoising-based methods in the context of today's self-supervised learning research.

**Limitations.** Our work studies DDM and DAE empirically, with limited theoretical analysis; while we have locally shuffled the deconstruction order and verified our major claims, it's impossible to experiment all the orders; and while *l*-DAE outshines prior baselines in certain settings, it falls short in others. Nonetheless, we hope our study provides clean and valuable signals to the community, which can help guide future research along this direction.

**Acknowledgements.** We thank Pascal Vincent, Mike Rabbat, and Ross Girshick for their discussion and feedback.

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

## A   IMPLEMENTATION DETAILS

**DiT architecture.** We follow the DiT architecture design (Peebles & Xie, 2023). The DiT architecture is similar to the original ViT (Dosovitskiy et al., 2021), with extra modifications made for conditioning. Each Transformer block accepts an embedding network (a two-layer MLP) conditioned on the time step $t$. The output of this embedding network determines the scale and bias parameters of LayerNorm (Ba et al., 2016), referred to as adaLN (Peebles & Xie, 2023). Slightly different from Peebles & Xie (2023), we set the hidden dimension of this MLP as 1/4 of its original dimension, which helps reduce model sizes and save memory, at no accuracy cost.

**Training.** The original DiTs (Peebles & Xie, 2023) are trained with a batch size of 256. To speed up our exploration, we increase the batch size to 2048. We perform linear learning rate warm up (Goyal et al., 2017) for 100 epochs and then decay it following a half-cycle cosine schedule. We use a base learning rate $blr$ = 1e-4 (Peebles & Xie, 2023) by default, and set the actual $lr$ following the linear scaling rule (Goyal et al., 2017): $blr \times$ batch_size / 256. No weight decay is used (Peebles & Xie, 2023). We train for 400 epochs by default. On a 256-core TPU-v3 pod, training DiT-L takes 12 hours.

**Linear probing.** Our linear probing implementation follows the practice of MAE (He et al., 2022). We use *clean*, 256×256-sized images for linear probing training and evaluation. The ViT output feature map is globally pooled by average pooling. It is then processed by a parameter-free Batch-Norm (Ioffe & Szegedy, 2015) layer and a linear classifier layer, following He et al. (2022). The training batch size is 16384, learning rate is $6.4 \times 10^{-3}$ (cosine decay schedule), weight decay is 0, and training length is 90 epochs. Randomly resized crop and flipping are used during training and a single center crop is used for testing. Top-1 accuracy is reported.

While the model is conditioned on $t$ in self-supervised pre-training, conditioning is not needed in transfer learning (*e.g.*, linear probing). We fix the time step $t$ value in our linear probing training and evaluation. The influence of different $t$ values (out of 1000 time steps) is shown as follows:

| fixed $t$ | 0 | 10 | 20 | 40 | 80 |
|---|---|---|---|---|---|
| w/ clean input | 64.1 | 64.5 | 64.1 | 63.3 | 62.2 |
| w/ noisy input | 64.2 | 65.0 | 65.0 | 65.0 | 64.5 |

We note that the $t$ value determines: (i) the model weights, which are conditioned on $t$, and (ii) the noise added in transfer learning, using the same level of $t$. Both are shown in this table. We use $t$ = 10 and clean input in all our experiments, except Tab. 5 where we use the optimal setting.

Fixing $t$ also means that the $t$-dependent MLP layers, which are used for conditioning, are not exposed in transfer learning, because they can be merged given the fixed $t$. As such, our model has the number of parameters just similar to the standard ViT (Dosovitskiy et al., 2021), as reported in Tab. 5.

The DiT-L (Peebles & Xie, 2023) has 24 blocks where the first 12 blocks are referred to as the "encoder" (hence ViT-$\frac{1}{2}$L) and the others the "decoder". This separation of the encoder and decoder is artificial. In the following table, we show the linear probing results using different numbers of blocks in the encoder, using the same pre-trained model:

| enc. blocks | 9 | 10 | 11 | 12 | 13 | 14 | 15 |
|---|---|---|---|---|---|---|---|
| acc. | 58.5 | 62.0 | 64.1 | **64.5** | 63.6 | 61.9 | 59.7 |

The optimal accuracy is achieved when the encoder and decoder have the same depth. This behavior is different from MAE's (He et al., 2022), whose encoder and decoder are asymmetric.

**End-to-end fine-tuning.** We closely followed MAE's protocol (He et al., 2022). We use clean, 256×256-sized images as the inputs to the encoder. Globally average pooled outputs are used as features for classification. The training batch size is 1024, initial learning rate is $4 \times 10^{-3}$, weight decay is 0.05, drop path (Huang et al., 2016) is 0.1, and training length is 100 epochs. We use a layer-wise learning rate decay of 0.85 (B) or 0.65 (L). MixUp (Zhang et al., 2018a) (0.8), CutMix (Yun et al., 2019) (1.0), RandAug (Cubuk et al., 2020) (9, 0.5), and exponential moving average (0.9999) are used, similar to He et al. (2022).

**Object detection and segmentation transfer.** We adopted the COCO recipe from ViTDet (Li et al., 2022), except: window attention size is set to $8{\times}8$, as the pre-training size for *l*-DAE is $256{\times}256$; a layer-wise learning rate decay of 0.95; a drop path of 0.1 (B) or 0.2 (L).

## B   VISUALIZATIONS OF THE LATENT SPACE

Thanks to the simplicity of using patches, for the three patch-wise tokenizers (VAE/AE/PCA), we can visualize their filters in the patch space. We show below in Fig. 8.

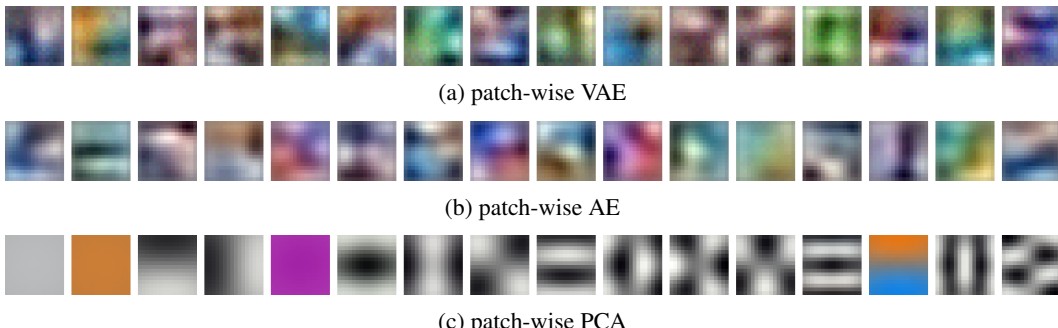

(a) patch-wise VAE

(b) patch-wise AE

(c) patch-wise PCA

Figure 8: **Visualization of the patch-wise tokenizer**. Each filter corresponds to a row of the linear projection matrix $V$ ($d{\times}D$), reshaped to $16{\times}16{\times}3$ for visualization. Here $d{=}16$.

## C   OBJECT SEGMENTATION RESULTS

Besides Average Precision (AP) for bounding box detection, the evaluation of ViTDet (Li et al., 2022) also includes results on object segmentation detection. We list them in Tab. 6 below.

| method | ViT-B | | ViT-L | |
|---|---|---|---|---|
| | $AP^{box}$ | $AP^{mask}$ | $AP^{box}$ | $AP^{mask}$ |
| Supervised | 47.6 | 42.4 | 49.6 | 43.8 |
| MAE | 51.2 | 45.5 | 54.6 | 48.6 |
| *l*-DAE | 51.6 | 45.8 | 54.4 | 48.2 |

Table 6: **Object detection and segmentation results** on COCO using ViTDet (Li et al., 2022). *l*-DAE performs similarly to MAE and outperforms supervised pre-training.

