# OpenReview forum: "Deconstructing Denoising Diffusion Models for Self-Supervised Learning"
_ICLR.cc/2025/Conference — ICLR 2025 Poster_

### Official Review · Reviewer_NrMn · 2024-11-03

**Soundness:** 3
**Presentation:** 3
**Contribution:** 3
**Rating:** 6
**Confidence:** 3

**Summary:**

In this paper, the authors perform an extensive ablation study on modern Denoising Diffusion Models (DDMs) to identify the critical components for effective representation learning, comparing these models to traditional denoising autoencoders (DAEs). The authors begin by comparing DDMs and DAEs on their design and purpose, indicating that generation (DDM objective) is not directly connected with good representations (DAE objective), and suggesting a possible trade-off between these them.

Then the authors test various modifications to make DDMs more similar to DAEs while preserving high-quality representations. Notably, they find that the tokenizer is beneficial mainly for its dimensionality reduction role and that a simplified approach, like patch-wise PCA, can serve this function without compromising performance. They also identify less critical components: noise scheduling (analogous to data augmentation), class conditioning (which may lessen the model’s need to capture class-specific, fine-grained semantics), and predicting noise versus the original image.

Based on these findings, the authors propose **l-DAE**, a DAE that operates in the latent domain. They validate the effectiveness of l-DAE through experiments on classification and object detection tasks, showing its competitive performance.

**Strengths:**

1. The authors aim to bridge modern self-supervised learning with classical methods by deconstructing diffusion denoising models (DDMs) and propose a simple but effective representation learner similar to traditional denoising autoencoders (DAEs).

2. Through this novel deconstructing process, the authors provide key insights. For instance, they highlight the importance of the low-dimensional latent space into which the tokenizer maps the patches, as it plays a crucial role in learning robust representations. Additionally, they demonstrate that adding noise in the latent space is more effective than in the pixel space, a finding that invites further exploration of latent space structures.

**Weaknesses:**

1. In the paper, the authors make numerous claims but test them only on a specific task (ImageNet classification) and model (DiT-L). Furthermore, the deconstructing process faces a limitation: the components may be correlated, making a sequential analysis potentially inadequate.
2. Some of the claims may be overlooked. For instance, the statement that 'multiple levels of noise is analogous to a form of data augmentation' (lines 416–418) may be overly simplified. Prior research (https://arxiv.org/abs/1406.3269) has shown that combining representations at different noise levels can lead to significant improvements."

**Questions:**

1. Could you explain on why noise scheduling can be considered a form of data augmentation? Is there any ablation study showing that the effects of noise scheduling and data augmentation are comparable?
2. Regarding the section on predicting the original image (Lines 392–406), shouldn't the matrix $V$ be of size $D \times d$? Also, if the weights $w_i$ are all 1, the loss would again become the reconstruction loss on the latent space. This suggests that the relative scale of these weights is important. Do you have any ablation studies on this aspect?"

---

> ### Author Response · Authors · 2024-11-24
>
> > In the paper, the authors make numerous claims but test them only on a specific task (ImageNet classification) and model (DiT-L).
>
> Thank you for bringing up this point. We would like to clarify that the effectiveness of our final l-DAE—featuring the patch-PCA-based tokenizer and associated design choices—extends well beyond ImageNet classification and the DiT-L model. For instance, we observed that using larger models (with ViT-L encoder and decoder) results in a significant performance improvement of 5.9% (Table 4c). Additionally, when transferring to object detection tasks with ViT-B, l-DAE outperforms MAE on COCO. These findings bolster our confidence in the generalizability of our approach.
>
> > Furthermore, the deconstructing process faces a limitation: the components may be correlated, making a sequential analysis potentially inadequate.
>
> Thank you for highlighting this important point. We acknowledge that the components of the deconstructing process may be interdependent, and it is indeed infeasible to experiment with all possible orders. However, we did shuffle the order of certain steps locally during our exploration. For instance, we replaced the noise schedule before removing VQGAN losses (Table 1) and revised noise prediction in our final l-DAE recipe (footnote 4). Despite these variations, the overall takeaway remains consistent, giving us confidence in our assessment within reasonable limits.
>
> > Some of the claims may be overlooked. For instance, the statement that 'multiple levels of noise is analogous to a form of data augmentation' (lines 416–418) may be overly simplified. Prior research (https://arxiv.org/abs/1406.3269) has shown that combining representations at different noise levels can lead to significant improvements."
>
> Thanks for the reference – this is definitely related to our claim about multiple levels of noise and we will discuss it. Upon checking scheduled denoising autoencoders (ScheDA), we note some key differences. Specifically, it proposes to *sequentially* reduce the noise levels as the training proceeds, and end training with low noise levels, which is close in distribution to the original data distribution.
>
> In contrast, DDMs train across multiple noise levels *simultaneously*, a design intended to support the diffusion-based generation process, which requires the model to operate effectively with all noise levels after training. While both approaches leverage multiple noise levels, their motivations, designs, and goals are quite different. ScheDA proposes to vary noise level for helping representation learning, whereas DDMs are tailored to facilitate generation from pure noise.
>
> We appreciate the opportunity to refine our claim and will consider incorporating a discussion of ScheDA to provide a more balanced perspective on the use of multiple noise levels. Thank you again for pointing this out.
>
> > Could you explain why noise scheduling can be considered a form of data augmentation? Is there any ablation study showing that the effects of noise scheduling and data augmentation are comparable?
>
> Thank you for the question. The explanation is as follows: a “noised” image can be considered a perturbation of the clean image that preserves its high-level semantics (so that the label remains unchanged) while altering the low-level details. In this sense, it creates a new version of the original image, effectively augmenting the data distribution. By training models on this augmented distribution, they can potentially generalize better and become more robust to noise-related variations.
>
> It is important to clarify that we do not claim adding noise is equivalent to a *specific* form of data augmentation. Rather, conceptually, adding noise can be understood as a *new* form of data augmentation different from the popular image augmentations used today, due to its ability to introduce variability in the data while preserving semantic content. We appreciate the opportunity to elaborate on this and hope it clarifies our perspective.

---

> ### Author Response · Authors · 2024-11-24
>
> > Regarding the section on predicting the original image (Lines 392–406), shouldn't the matrix
> V be of size Dxd? Also, if the weights are all 1, the loss would again become the reconstruction loss on the latent space. This suggests that the relative scale of these weights is important. Do you have any ablation studies on this aspect?
>
> Thank you for carefully checking the technical details.
> - Regarding the size of $V$: it represents the full PCA basis and thus has dimensions  $D \times D$, with no basis dropped for dimensionality reduction.
> - As for the case where all the weights are set to 1, the intrinsic dimensionality of the input to the autoencoder remains $d$. Noise is specifically added to the first $d$ principal components before projecting them back to $D$ dimensions. This differs from directly adding noise to the $D$-dimensional input or to all $D$ principal components. Consequently, predicting the original image is not merely a *reconstruction* task — it also requires the model to infer the remaining  $D-d$  dimensional information (the residue) based on the noised input of intrinsic dimension $d$. This additional task is distinct from the first, justifying the use of different loss weights.
> - We conducted a search for the per-dimensional loss weight for the residue, and found 0.1 to be the best-performing value:
> | per-dimensional loss | acc |
> |---------|--------|
> | 0.01 |  63.7  |
> | 0.03 |  63.9  |
> | 0.1   |  **64.5**  |
> | 0.3   |  63.8  |
> | 1.0   |  61.5  |
>
> We hope this clarifies the setup and reasoning behind the loss weighting. Please let us know if further details are needed!

---

> > ### Comment · Reviewer_NrMn · 2024-11-27
> >
> > I thank the authors for their efforts in addressing my concerns. Regarding Points 1 and 2, I acknowledge the effectiveness of l-DAE and find many of the authors' insights compelling. However, to me, these findings do not provide a sufficiently robust endorsement of the deconstruction process.
> >
> > For Points 3 and 4, while I still believe that the noise level and schedule may play a critical role in general representation learning, the authors’ response—along with their findings demonstrating the effectiveness and robustness of using $\sigma = \sqrt{1/3}$—adequately addresses my concerns.
> >
> > Overall, I find the paper comprehensive and have decided to maintain my current rating.

---

> > > ### Author Response · Authors · 2024-11-27
> > >
> > > Thanks for the detailed thoughts and acknowledgement NrMn!

---

### Official Review · Reviewer_Lm16 · 2024-11-03

**Soundness:** 3
**Presentation:** 2
**Contribution:** 2
**Rating:** 8
**Confidence:** 4

**Summary:**

The paper studies the representation ability of generative denoising diffusion models (DDM). It specifically aims to identify crucial components for DDMs' representation ability during the process of removing modern components in DDMs until it becomes a simpler model very similar to classic Denoising Autoencoder (DAE). Notably, DAE is originally proposed for representation learning. The paper is highly empirical, conducting various experiments on different components, such as different loss terms, different tokenizers, class-conditioning, noise schedule, whether to predict clean data, etc. It can remove many components designed for generation, and show high-level representation abilities are not strictly related to generation ability.

**Strengths:**

(1) The paper has identified a thorough set of different components between classical DAE and modern diffusion models.

(2) The paper sheds light on a potential framework for understanding the representation ability of diffusion models - DAE.

(3) The paper is the first work trying to identify key representation components from generative models and could inspire future works.

(4) The paper has some interesting findings, such as the low-rank tokenizer is important, and the high-level representation ability may not correlate with the generation ability.

**Weaknesses:**

(1) The paper only presents empirical findings, with no theoretical analysis or practical applications.

(2) The complex possible choices of components make the experiment order not strictly natural and logical.

(3) Missing experiments on some possible choices of components make the conclusions of the paper not that strong, for example, it's hard to conclude whether predicting clean images is more helpful than predicting noise for representation learning.

**Questions:**

1. L 247: The paper claims "self-supervised learning performance is not correlated to generation quality". However, the selected tasks such as linear probes only consider coarse representations that are useful for high-level tasks. What about more fined representations such as segmentation, and positions of a specific object, etc, would it be correlated to generation quality?
2. L 369: why the DAE is expected to work directly on the image space, could you please explain the importance of working on the image space?

---

> ### Author Response · Authors · 2024-11-24
>
> > The paper only presents empirical findings, with no theoretical analysis or practical applications.
>
> Thanks for the comments. We acknowledge the lack of theoretical analysis as a limitation, which we have explicitly stated in the final paragraph of our paper (page 10). Regarding practical applications, while we do not explore specific end-use cases (e.g., self-driving cars or robotic manipulation), we believe that image classification and object detection serve as standard benchmarks for evaluating representation quality, and achieving strong performance on these tasks provides a robust foundation and stepping stone for such practical applications.
>
> > The complex possible choices of components make the experiment order not strictly natural and logical.
>
> Thank you for raising this point. We agree that modern DDMs are inherently complex, with multiple interconnected components, which is precisely why we adopted a deconstructive philosophy.
>
> The notion of whether the order is “strictly natural and logical” can be subjective. However, the order we chose largely reflects the natural trajectory of our research. This is summarized in the subsection titles of Section 4. First and foremost, we want to legitimize the pipeline for SSL; then we take a deep dive, looking for key factors that underlines the performance; and finally we align the full pipeline to classical DAE.
>
> Additionally, we have shuffled the experimental order locally to validate the robustness of our findings. For example, replacing the noise schedule before removing VQGAN losses (Table 1), or revising noise prediction in our final l-DAE recipe (footnote 4). The overall take-away remains the same -- therefore we are confident in our assessment.
>
> That said, it is impractical to explore all possible orders for deconstruction, and we acknowledge this as a limitation, explicitly stated as the second limitation on page 10. We hope this clarifies more, and we appreciate your understanding of the challenges in such a study.
>
> > Missing experiments on some possible choices of components make the conclusions of the paper not that strong, for example, it's hard to conclude whether predicting clean images is more helpful than predicting noise for representation learning.
>
> Thank you for your thoughtful feedback. We agree that there may be unresolved questions regarding specific design choices in our final pipeline. While we acknowledge that the pipeline is not perfect and there is certainly room for improvement, we would like to clarify the following:
>
> 1. Our primary goal in Section 4.3 was to preserve the representation quality of the patch-PCA-based DDM established in Section 4.2, while aligning the design as closely as possible to a classical DAE. As such, evaluating whether a specific choice, like predicting clean images versus predicting noise was secondary to this goal.
> 2. Regarding the specific question, we believe that according to [1], what matters underneath is the loss weighting function $\lambda_t$. We believe with a proper $\lambda_t$, both predicting clean images and predicting noise can achieve high representation quality.
> We hope this explanation clarifies our approach, and we appreciate your insights on these potential improvements.
>
> [1] Tim Salimans and Jonathan Ho. Progressive distillation for fast sampling of diffusion models. In ICLR, 2022.

---

> ### Author Response · Authors · 2024-11-24
>
> > L 247: The paper claims "self-supervised learning performance is not correlated to generation quality". However, the selected tasks such as linear probes only consider coarse representations that are useful for high-level tasks. What about more finer representations such as segmentation, and positions of a specific object, etc, would it be correlated to generation quality?
>
> > L 369: why the DAE is expected to work directly on the image space, could you please explain the importance of working on the image space?
>
> Thanks for asking these two great questions. We would like to address them jointly, starting from the second one.
> - The main motivation for ensuring that the model works directly on the image space is to make it fully compatible with downstream pipelines. For example, most standard object detectors and segmentation algorithms operate on the image space, not the latent token space, which may miss fine-grained details essential for such tasks.
> - This compatibility allows us to feed the l-DAE representations directly into ViTDet [2] (note that we will skip the per-patch PCA for ViTDet and directly feed images to the l-DAE initialized weights), enabling meaningful comparisons with other self-supervised learning methods on downstream tasks like object detection and segmentation.
> - Regarding the first question (L247), our focus has been on classification, which aligns with prior practices [3] that evaluate representations based on latent tokens. This is where we derived the conclusion that self-supervised learning performance, as measured by classification, is not correlated with generation quality.
> - Measuring beyond classification is important, but given the answer to the second question, we also want to point out that transferring tokenized representations to tasks such as object detection at that stage would be highly non-trivial. Attempting to do so would likely result in non-standard pipelines that may obscure meaningful insights. Thus, while your question is highly relevant, addressing it would require further research, which we hope can be explored in the future.
> - In light of this feedback, we will revise the statement at L247 to: “*Self-supervised learning performance measured by classification is not correlated with generation quality.*” We hope this clarifies our approach and appreciate your understanding.
>
> [2] Yanghao Li, Hanzi Mao, Ross Girshick, and Kaiming He. Exploring plain vision transformer
> backbones for object detection. In ECCV, 2022.
>
> [3] Tianhong Li, Huiwen Chang, Shlok Mishra, Han Zhang, Dina Katabi, and Dilip Krishnan. Mage: Masked generative encoder to unify representation learning and image synthesis. In CVPR. 2023.

---

> > ### Comment · Reviewer_Lm16 · 2024-11-25
> > **Thanks for the response**
> >
> > Dear authors, thanks for the clarifications. My questions have been resolved and I would like to raise my score.

---

> > > ### Author Response · Authors · 2024-11-25
> > >
> > > Thanks Lm16 for the feedback and acknowledgment!

---

### Official Review · Reviewer_SR52 · 2024-11-04

**Soundness:** 3
**Presentation:** 3
**Contribution:** 3
**Rating:** 6
**Confidence:** 4

**Summary:**

This paper studies the representation learning ability of denoising diffusion models. Through a set of ablation studies that deconstruct a denoising diffusion model into a classical denoising autoencoder (DAE), the authors observe that only a very few modern components (such as a low-dimensional latent space) are critical for learning good representations. Experiments also show that a latent DAE, which largely resembles the classical DAE, can perform competitively in self-supervised learning.

**Strengths:**

Studying the representation learning ability of denoising diffusion models is important and could provide useful insights for unifying models for generation and discrimination.

Overall, the paper is very well written. The experiments are solid, and the message is clear. The observation that a low-dimensional latent space is a critical component for representation learning is useful.

**Weaknesses:**

1. While the study begins with denoising diffusion models, it ultimately leads to models that demonstrate strong representations for classification but not for generation. The FID is reported only in Table 1, which reveals a significant contradiction between classification accuracy and FID.

3. For the goal of representation learning for classification without fine-tuning, the obtained latent DAE achieves slightly worse performance than MAE and contrastive learning.

4. The representation is extracted from the middle layer of the transformer for linear probing. Previous studies have found that the middle layer may not provide the best representation of a diffusion model for classification.

**Questions:**

1. What are the FID scores for other modifications beyond Table 1, such as operating in the image space with PCA?

2. Does better classification accuracy always lead to worse FID scores? In other words, are the tasks of generation and representation learning (for recognition) fundamentally contradictory to each other? Or could we unify them?

3. Why is the middle layer of the transformer chosen as the representation for linear probing?

---

> ### Author Response · Authors · 2024-11-24
>
> > The FID is reported only in Table 1, which reveals a significant contradiction between classification accuracy and FID.
>
> > What are the FID scores for other modifications beyond Table 1, such as operating in the image space with PCA?
>
> Thank you for raising this point. At the end of Table 1, the generation FID reaches 93.2—a level that we believe is no longer meaningful to report FIDs further, especially since state-of-the-art models typically achieve FIDs below 2. This aligns with our goal outlined in Section 4.1 to reorient DDMs specifically for self-supervised learning. After this reorientation, we focused exclusively on evaluating representation quality and did not assess generation quality further. We hope this explanation clarifies our approach, but we are happy to discuss this further if needed.
>
> > For the goal of representation learning for classification without fine-tuning, the obtained latent DAE achieves slightly worse performance than MAE and contrastive learning.
>
> Thank you for pointing this out. We fully acknowledge the remaining performance gap between l-DAE and MAE/contrastive methods for representation learning without fine-tuning. However, we would like to highlight that this gap has been *significantly narrowed* through our deconstructive process—from approximately 20% in Figure 4 to over 70% in Table 5a. Additionally, we ensured that all comparisons were conducted *fairly*, without employing any extra techniques to artificially boost l-DAE’s performance. This reflects our commitment to providing clear and unbiased takeaways for readers. We hope this addresses your concern, and we are happy to elaborate further if needed.
>
> > The representation is extracted from the middle layer of the transformer for linear probing. Previous studies have found that the middle layer may not provide the best representation of a diffusion model for classification.
>
> > Why is the middle layer of the transformer chosen as the representation for linear probing?
>
> Thank you for your comment; this is indeed a valid concern. We investigated “which layer to take” from the pre-trained l-DAE for linear probing, as detailed in the table at L744–747. In our setup, we observed that the middle layer (the 12th layer in a 24-layer ViT-L) provided the best linear probing performance. We hope this clarifies our choice.
>
> > Does better classification accuracy always lead to worse FID scores? In other words, are the tasks of generation and representation learning (for recognition) fundamentally contradictory to each other? Or could we unify them?
>
> Thank you for this question. In our study, we observed that representation quality (as measured by linear probe accuracy on ImageNet) does not correlate well with generation quality, and there is indeed a clear trade-off between the two. Certain design choices are more favorable for generation, while others enhance representation learning. While the idea of a unified framework is both intuitively appealing and practically desirable, achieving this balance remains an open challenge and an exciting direction for future research.

---

> > ### Comment · Reviewer_SR52 · 2024-11-27
> >
> > I appreciate the authors' detailed responses, which have addressed my questions. Thank you for sharing your thoughts on the trade-off between representation quality and generation quality. I maintain a positive rating.

---

> > > ### Author Response · Authors · 2024-11-27
> > >
> > > Thanks for the acknowledgement SR52!

---

### Official Review · Reviewer_Lfx6 · 2024-11-06

**Soundness:** 3
**Presentation:** 3
**Contribution:** 3
**Rating:** 6
**Confidence:** 2

**Summary:**

In this paper, the author studied the representation learning abilities of denoising-autoencoder-based diffusion models (DDMs). Throughout extensive ablation studies, they explored how various components of modern DDMs influence self-supervised representation learning. At the core of their philosophy is to deconstruct a DDM, changing it step-by-step into a classical DAE. This research process demonstrates that the main critical component for a DAE to learn good representations is a tokenizer that creates a low-dimensional latent space.

**Strengths:**

**1.** In general, this paper contributes significantly to the intersections of diffusion models and representation learning.  The findings open up new avenues for future research in leveraging diffusion processes to enhance representation quality across diverse applications

**2.** The authors conducted extensive experiments to support their results. The paper is well-written and easy to follow.

**Weaknesses:**

The reviewer has the following major concerns about this paper:

**1.** It is not comprehensive to study the representation ability of diffusion models only by considering the classification of downstream tasks. The authors should provide more experiments on other tasks to support their conclusions.

**2.** Although the observations of this work are really new and interesting, the authors seem to not fully discuss the implications of these findings.

**Questions:**

**1.** The authors mainly focused on investigating the representation learning abilities of DiTs. Are there similar observations on U-Net-based diffusion models?

**2.** Based on the experimental results, can we conclude that adding noise primarily impacts the generation capabilities of diffusion models rather than their representation learning ability? Are there any insights for this observation?

---

> ### Author Response · Authors · 2024-11-24
>
> > It is not comprehensive to study the representation ability of diffusion models only by considering the classification of downstream tasks
>
> Thank you for the valuable feedback. We agree that evaluating the representation ability beyond classification is important. In the submission, we have included results for transferring our final l-DAE to COCO object detection and segmentation, as shown in Table 5c and Table 6. The key takeaway is consistent with the results from fine-tuned ImageNet classification: l-DAE outperforms MAE when using ViT-B, while it underperforms with ViT-L. Notably, both autoencoding-based methods significantly outperform supervised pre-training.
>
> > Although the observations of this work are really new and interesting, the authors seem to not fully discuss the implications of these findings.
>
> Thanks for the feedback. Two broader implications of our findings are: 1) l-DAE as a simplification of the modern DDM can serve as an alternative to existing SSL methods for representation learning, with different properties and behaviors; 2) Since l-DAE is close to DDM, an especially interesting next step would be to reconcile the trade-offs between representation learning and generative learning, and build a truly unified model.
>
> > The authors mainly focused on investigating the representation learning abilities of DiTs. Are there similar observations on U-Net-based diffusion models?
>
> Thank you for your question. We selected DiT for this study primarily because its pre-trained representations are standard ViTs, which facilitate straightforward transfer to downstream tasks such as object detection. More importantly, this choice ensures fair comparisons with other pre-training methods like MoCo and MAE. As a result, we have not yet conducted experiments on U-Net-based diffusion models. However, we are optimistic that the importance of a low-dimensional latent space will generalize to architectures beyond DiTs, and potentially study it as part of our future exploration.
>
> > Based on the experimental results, can we conclude that adding noise primarily impacts the generation capabilities of diffusion models rather than their representation learning ability? Are there any insights for this observation?
>
> Thank you for this question. The short answer is no. Based on our experimental results, we conclude that adding noise (and the associated process of “denoising”) influences both generation and representation learning. In contrast, “diffusion modeling” (and the associated noise schedules) appears to primarily impact the generative capabilities of the model. This distinction is why our final approach is named l-DAE, with the “D” for “denoising” rather than “diffusion”. We hope this clarification provides useful insight into the separation of these processes and their respective roles.

---

> > ### Comment · Reviewer_Lfx6 · 2024-11-26
> >
> > Dear authors,
> >
> > Thanks for the clarifications. I will keep the score unchanged.

---

> > > ### Author Response · Authors · 2024-11-26
> > >
> > > Thanks for the acknowledgement Lfx6!

---

### Author Response · Authors · 2024-11-24
**Shared response from authors**

We sincerely thank all the reviewers for their time, efforts, and thoughtful feedback. We are delighted to see all the reviews are positive about our work, with remarks highlighting various aspects:

- **Novelty**: “the observations of this work are really new and interesting” (Lfx6), “the paper is the first work trying to identify key representation components from generative models and could inspire future works” (Lm16), “interesting findings” (Lm16), “novel deconstructing process” (NrMn)
- **Writing**: “the paper is well-written and easy to follow” (Lfx6), “the paper is very well written … the message is clear” (SR52)
- **Significance**: “​​contributes significantly to the intersections of diffusion models and representation learning … open up new avenues for future research” (Lfx6), “Studying the representation learning ability … is important and could provide useful insights for unifying models for generation and discrimination” (SR52), “the paper sheds light on a potential framework for understanding the representation ability of diffusion models” (Lm16), “provide key insights” (NrMn)
- **Experiments**: “extensive ablation studies … conducted extensive experiments to support their results” (Lfx6), “the experiments are solid” (SR52), “highly empirical … conducting various experiments on different components” (Lm16), “extensive ablation study” (NrMn)

We have carefully addressed each reviewer’s comments and questions individually below, and we hope that our responses address all remaining concerns. Should there be any further clarifications needed, we would be happy to provide them.

---

### Meta-Review · Area_Chair_15za · 2024-12-20

**Metareview:**

In this paper the authors conduct a sequence of experiments which progressively transforms a Denoising Diffusion Model in a Denoising Autoencoder to understand how the various aspects of the model impact the overall performance and to identify which components are essential (or not) for the model.

Overall, the reviewers are largely in agreement that the paper makes an interesting contribution.  While several weaknesses are noted by the reviewers, none of the issues appear to be critical flaws and instead appear to be more of avenues for potential improvement of the work, and I believe this work would be of interest to the community.

**Additional Comments On Reviewer Discussion:**

The authors were largely responsive to the questions raised by the reviewers, with one reviewer raising their score.

---

### Decision · Program_Chairs · 2025-01-22

Accept (Poster)